# Severity of Anaemia Has Corresponding Effects on Coagulation Parameters of Sickle Cell Disease Patients

**DOI:** 10.3390/diseases7040059

**Published:** 2019-12-17

**Authors:** Samuel Antwi-Baffour, Ransford Kyeremeh, Lawrence Annison

**Affiliations:** 1Department of Medical Laboratory Sciences, School of Allied Health Sciences, College of Health Sciences, University of Ghana, P.O. Box KB 143 Accra, Ghana; rkyeremeh@gmail.com; 2Department of Medical Laboratory Sciences, School of Allied Health Sciences, Narh-Bita College, P.O. Box Co1061 Tema, Ghana; larryannison@yahoo.com

**Keywords:** haemoglobin, sickle cell disease, platelets, prothrombin, coagulation, anaemia, antithrombin, haemolysis

## Abstract

Sickle cell disease (SCD) is an inherited condition characterized by chronic haemolytic anaemia. SCD is associated with moderate to severe anaemia, hypercoagulable state and inconsistent platelet count and function. However, studies have yielded conflicting results with regards to the effect of anaemia on coagulation in SCD. The purpose of this study was to determine the effect of anaemia severity on selected coagulation parameters of SCD patients. Four millilitres of venous blood samples were taken from the participants (SCD and non-SCD patients) and used for analysis of full blood count and coagulation parameters. Data was analysed using SPSS version-16. From the results, it was seen that individuals with SCD had a prolonged mean PT, APTT and high platelet count compared to the controls. There was also significant difference in the mean PT (p = 0.039), APTT (p = 0.041) and platelet count (p = 0.010) in HbSS participants with severe anaemia. Mean APTT also showed significant difference (p = 0.044) with severe anaemia in HbSC participants. It can be concluded that SCD patients have prolonged PT, APTT and increased platelet count which might predispose them to bleeding episodes and thrombocytosis. Significant difference was also seen between severity of anaemia and mean PT, APTT and platelet count in HbSS individuals.

## 1. Introduction

Sickle cell disease (SCD) is a hereditary blood disorder characterized by chronic haemolytic anaemia. The clinical manifestations arise from the tendency of abnormal Haemoglobin-S (HbS) to polymerize and deform red blood cells (RBCs) into a characteristic sickle shape [1]. It is an autosomal recessive disease caused by a point mutation in haemoglobin (Hb) by the substitution of valine for glutamic acid at position 6 of the beta (β)–globin chain found on chromosome 11 [2]. Sickle cell disease denotes all entities associated with sickling of Hb and is categorized into the homozygous form Haemoglobin SS (HbSS) also known as sickle cell anaemia (SCA) and the compound heterozygous forms such as Haemoglobin SC (HbSC) and Haemoglobin S/Beta-thalassemia (HbSβ) [3]. Some SCD disease patients alternate between steady state and episodes of crises. A patient with SCD is said to be in a steady state when the person is free of pain, infection, or other disease processes [4]. Crisis refers to episodes of acute illness attributable to the sickling phenomenon in which there is a sudden worsening of symptoms and signs in patients with SCD who were previously in steady state [5]. Sickle cell disease is associated with moderate to severe anaemia, hypercoagulable state and has also been found to have an effect on platelet count and function [6,7,8].

Coagulation is a complex network of interactions involving the blood vessels, platelets and clotting factors and is initiated by intrinsic and extrinsic enzymatic pathways which ultimately lead to fibrin formation [9]. Activated partial thromboplastin time (APTT), prothrombin time (PT), thrombin time (TT) and fibrinogen concentration are routine coagulation tests used to assess pathological alterations of haemostatic and coagulation systems to guide clinical therapy along with the platelet count [10,11]. Platelets are circulating anuclear cellular fragments, about 2–3 µm in diameter produced from the fragmentation of megakaryocytes in the bone marrow [12]. The main function of platelets is stopping of bleeding followed by vascular injury and maintain haemostasis. Platelet count is done to determine the amount of platelet in the blood of an individual [13].

According to Sandra (2012) prothrombin time measures integrity of the extrinsic coagulation pathway and the common pathway (Factors V, X, prothrombin and fibrinogen) whilst APTT tests for deficiencies in the intrinsic pathway, specifically factors VIII, IX, XI, XIII and deficiencies in the common pathway [14]. Various factors can promote or inhibit coagulation. Protein S, protein C and antithrombin, referred to as the natural inhibitors of coagulation, inhibit specific clotting factors when activated after injury, providing a control mechanism which limits the amount of clot formed [15]. Plasma haemoglobin and its breakdown product haem can directly activate endothelial cells and promote coagulation [16]. Moreover, Ataga and Key (2007) stated that exposure of Phosphatidylserine (PS) on the surface membrane of RBCs promotes coagulation activation and hence hypercoagulability [17].

Sickle cell disease is associated with hypercoagulable state that may contribute to certain morbidities seen in the disease [18]. Platelets, procoagulants, anticoagulants and fibrinolytic system which are components of haemostasis are altered in the disease contributing to the morbidity [17]. A study done by Chinawa et al. (2013) suggests that activation of coagulation present in SCD is partly due to haemolysis with resultant scavenging of nitric oxide (NO) by cell free plasma haemoglobin released during haemolysis [19]. Wright et al. (1997) also observed in their study decreased levels of natural anticoagulant, protein C and protein S in their study [20]. These reduced levels may be the consequence of chronic consumption of coagulation factors arising from increased thrombin generation and haemolysis which occur in the vascular endothelium and which can enhance anaemia in SCD patients [21].

Anaemia is classified as mild, moderate or severe based on the concentrations of Hb in the blood. According to WHO (2011), mild anaemia corresponds to Hb concentration level of 11.0–11.9 g/dL (women) and 11.0–12.9 g/dL (men) with moderate anaemia having Hb concentration of 8.0–10.9 g/dL (for both male and female) and severe anaemia with level less than 8.0 g/dL (for both male and female) [22]. Individuals with SCD do not have anaemia at birth due to high level of haemoglobin F but develop chronic haemolytic anaemia with increased synthesis of adult Hb with acute episodes of reduction in haemoglobin (anaemic crises) throughout life [23].

According to Mehta et al. (2012), anaemia may require more blood flow to the brain to compensate for the lack of oxygen [24]. However, this increase in blood flow can cause endothelial vessel damage and can lead to platelet activation causing a cascade of thrombus formation to occur [24]. Ataga et al. (2012) in their study observed hypercoagulability when blood was diluted 30% with saline [25]. Moreover, a study done by Scharbert et al. (2011) showed significantly increased velocity of platelet aggregation in anaemic blood samples and shortened clotting time [26]. Kumar et al. (2009) in their study on anaemia and Intra-Cerebral-Haemorrhage (ICH) showed that patients with anaemia had larger ICH volumes [27]. Thus, there are conflicting reports on the effect of anaemia on coagulation. While some say anaemia predisposes one to hypercoagulability, others say it predisposes an individual to excessive bleeding. Since SCD patients are known to show varying degrees of anaemia, it would be important to determine the effect of severity of anaemia on their coagulation pattern which is the aim of this study. The outcome will add to the knowledge on coagulopathy found in SCD based on the degree of anaemia and may help improve management of individuals with the disease.

## 2. Materials and Methods

### 2.1. Ethics Approval and Consent to Participate

All subjects gave their informed consent for inclusion before they participated in the study. The study was conducted in accordance with the Declaration of Helsinki, and the protocol was approved by the Ethics Committee of the School of Biomedical and Allied Health Sciences, University of Ghana (Project identification code: ET./10334520/AA/1A/2013-2014).

### 2.2. Participant Recruitment

During recruitment of the participants, it was ensured that no participant was on anticoagulant medications, had recent blood transfusions, had known liver disease that would affect production of clotting factors and had known renal failure that would affect haematocrit. Finally, individuals on medications such as hydroxyurea that could lower the platelet count were excluded from the study.

### 2.3. Sample Collection and Processing

Four millilitres (4 mL) of venous blood sample was taken from the antecubital fossa of the participants, of which 2.2 mL was dispensed into Ethylene-Diamine Tetra-acetic Acid (EDTA) tube and 1.8 mL into a sodium citrate tube and swirled gently to enable appropriate mixing of anticoagulant with blood. The blood samples were then transferred to the laboratory for analysis. 

### 2.4. Sample Analysis

The laboratory investigation carried out included full blood count for Hb and platelets, coagulation studies (APTT and PT) and Hb electrophoresis for confirmation of Hb genotype status. 

#### 2.4.1. Full Blood Count (FBC)

The FBC analysis was done using ABX Micros ES 60 Haematology Analyzer following the manufacturer’s principle (Horiba Medical, Kyoto, Japan). Haemoglobin concentration and platelet count was obtained from the FBC. 

#### 2.4.2. Coagulation Analyses

Activated Partial Thromboplastin Test (APTT) and Prothrombin Test (PT) were done using a coagulation auto-analyser (AC-2005, Guangzhou, China). Reagents used were manufactured by cypress diagnostics and the principle for the test obtained from an insert in the reagent.

##### Principle for APTT

The reagent is cephalin, a brain lipid extract that performs as a platelet substitute. Micronized silica is used as an activator of the factors XI and XII. When these reagents and calcium chloride are added to citrated plasma, the factors of the intrinsic coagulation pathway are activated; the time for the plasma to clot is then measured.

##### Principle for PT

When citrated plasma is recalcified in the presence of a high concentration of tissue factor reagent (tissue thromboplastin), the factors of the extrinsic coagulation pathway are activated; the time for the plasma to clot is then measured.

#### 2.4.3. Haemoglobin Electrophoresis 

Haemoglobin electrophoresis was done on the sample using cellulose acetate membrane at Ph 8.5. The principle and procedure for Hb electrophoresis were followed as previously described by Dacie and Lewis (2012) [28]. 

### 2.5. Data Analysis

Data was analysed using SPSS version 16 (New York, USA). Descriptive variables were presented in frequencies, percentages, means and standard deviation. ANOVA was used to check if there was a significant difference between the mean APTT, PT and platelet count of patients and controls. Moreover, to check if there was a significant difference between the various categories of Anaemia and mean PT, APTT and platelet count on patients with SCD. Post hoc analysis was done to ascertain the differences in means between the cases and controls. Significant was at P < 0.05.

## 3. Results

A total of two hundred and thirty-two (232) participants were used in this study out of which 161 formed the study group and 71 the control group. By way of gender, 112 (48.3%) were males and 120 (51.7%) females. The mean age was 23.63 ± 2.92 years for those with HbAA; 25.93 ± 10.96 years for HbSS and 35.56 ± 15.72 years for HbSC. Clinical variables obtained also indicated that a higher number of the HbSS participants showed pallor and also get pain crisis, fatigue, dactylitis and frequent infection than those with Hb SC and Hb AA (Table 1).

Out of the total number (161) of the cases, 144 fell within the Hb level of 8.0–11.9 that are seen as being anaemic. Specifically, those who had haemoglobin <8.0g/dL were said to have severe anaemia, 8.0–10.9 g/dL as moderate and 10.9–11.9g/dL as mild anaemia. Subsequently, it was seen that for HbSS, 49 had Hb <8.0g/dL (severe anaemia), 42 had Hb between 8.0–10.9g/dL (moderate anaemia) and 2 between 10.9–11.9g/dL (mild anaemia). For HbSC, 2 had Hb <8.0g/dL, 32 had between 8.0–10.9g/dL and 17 between 10.9–11.9g/dL. For the controls, only 7 participants had Hb level between 10.9–11.9g/dL and were categorized as having mild anaemia. The mean value for the PT and platelet results for HbSS participants was slightly higher than those obtained by HbSC and HbAA. However, with regards to APTT, participants with HbSC had a higher value than the HbSS and HbAA participants (Table 2).

The various categories of anaemia within the different genotypes of the cases were compared with the coagulation parameters. It was seen that there was a significant difference between the mean PT, APTT and platelet count and severe anaemia within participants with HbSS (p = 0.039, p = 0.041 and p = 0.01). There was also significance between the mean platelet count and moderate anaemia. However, no significant difference was seen when PT, APTT and platelet count results were compared with the other anaemia categories. Similar comparison was done with participants with HbSC and only APTT showed significant difference in participants with severe anaemia (Table 3).

When post-hoc analysis was done, the study showed significant differences in the mean levels of the parameters measured between the cases and controls. However, there was no significant mean difference in the APTT of patients with HbSS and HbSC (Table 4).

## 4. Discussion

This study was done to check the mean PT, APTT and platelet count of individuals with sickle cell disease (SCD–HbSS and HbSC) and without (HbAA) in relation to the degree of anaemia seen in them. The study comprised of 232 participants out of which 161 were patients with SCD and form the cases. These were further divided into 95 (m = 34, f = 61) HbSS and 66 (m = 25, f = 41) HbSC. The control group had 71 (m = 53, f = 18) participants with HbAA. Clinical findings indicated that a substantial number of those with HbSS had pallor appearance, get pain crisis, fatigue, dactylitis and frequent infections as compared with those with HbSC and HbAA. Again, the results obtained from the study found three categories of anaemia (severe, moderate and mild) among the participants. In this regard, a higher number of HbSS participants presented with severe anaemia more than the other study groups. Generally, participants with HbSS had lower Hb levels when compared to HbSC and HbAA participants. This was due to the chronic haemolysis, aplastic crisis as well as a possible blood loss in haematuria associated with SCD [29]. It therefore implies that the rate of RBC destruction is very high in SCD and hence contributes to the lower levels of RBC and other RBC indices including Hb.

With coagulopathy, the study found that individuals with SCD had a prolonged mean PT, APTT and high platelet count when compared to controls with normal haemoglobin (HbAA). The finding of prolonged PT and APTT among the SCD participants could be as a result of reported decrease in the plasma levels of factor V [30], total factor VII [31], and factor VII zymogen [31,32] in SCD patients. This was in line with a study by Buseri et al. (2006) that found the mean PT in HbSS patients to be significantly higher than the mean PT value in HbAA control subjects [33]. The mean platelet count in this study which was significantly higher in the SCD participants compared to the HbAA controls is in keeping with earlier reports of elevated platelet count in SCD, which was attributed to the auto-splenectomy and/or loss of splenic function that frequently occurs in SCD patients [34]. However, the auto-splenectomy and/or loss of splenic function is lower in HbSC patients compared to HbSS patients [35,36], and this may explain why the mean platelets in the HbSC participants was lower than that of HbSS participants but was higher than the mean counts for HbAA controls. This suggests that though splenic function may be better preserved in HbSC patients than HbSS patients, it was still impaired compared to HbAA controls.

The categories of anaemia detected were compared with the coagulation parameters measured and it was seen that there was significant difference in the mean PT (p = 0.039), mean APTT (p = 0.041) and mean platelets (p = 0.010) against severe anaemia among HbSS subjects. Only mean APTT showed significant difference (p = 0.044) with severe anaemia among HbSC participants. These findings are in keeping with findings by Chinawa et al. (2013) and Nilesh et al. (2014) who reported of a prolonged mean APTT and PT in HbSS patients when they were compared with controls with normal Haemoglobin [19,37]. Similarly, Raffini et al. (2006) also found a prolonged PT and APTT in SCD patients with HbSS in comparison to the controls with HbAA [38]. The increase in APTT and PT might have resulted from defective liver due to blockage by sickle cells as they traverse the capillary as seen in the study by Raffini et al. (2006) where they reported hepatic dysfunction amongst SCD patients. Since most of the coagulation factors are synthesized in the liver, a defective liver might result in decrease synthesis and as such may result in coagulopathy [38]. Decreased synthesis and increased consumption of coagulation factors was also observed by Wright et al. (1997) and Raffini et al. (2006) where the increased consumption was thought to be caused by the hypercoagulability seen in SCD patients [20,38].

The increase in platelet count observed in this study is also in line with the findings of the studies of Akinbami et al. (2012), Chinawa et al. (2013) and Francis (1991) where they collectively revealed an increase in platelet count when SCD patients with HbSS were compared with control with HbAA [7,19,39]. The increase in platelet count may be as a result of functional asplenia with a decrease in platelet pooling in the splenic population as postulated by Ataga and Orringer (2003) [40]. Another finding of the study was that the severity of anaemia had more effect on platelet count of patients with HbSS as significant difference was seen with both severe and moderate anaemia (p = 0.010 and 0.025, respectively). There was however no significant difference between the degrees of anaemia and platelets in patients with HbSC. This finding might have been as a result of difference in clinical severity seen in patients with HbSS as compared to patients with HbSC whereby the possible reduction of splenic sequestration of platelets or absence of spleen resulting in hyposplenism is more pronounced in HbSS than HbSC patients [34]. In this case, the decrease in platelets in HbSC individuals may not be as pronounced as that which will be seen in HbSS individuals. This fact is supported by a study by Westerman et al. (1999) where they established that significant differences in coagulation measurement between HbSS and HbSC diseases are consistent with differences in clinical severity between the diseases [41].

The study also found that 70.52% of patients with HbSS had their platelet count above the normal range as compared to 36.92% of patients with HbSC, meaning a higher percentage of HbSC participants fell into the normal range with regards to platelet count. Specifically, the HbSS participants with moderate anaemia had the highest number of platelets followed by those with severe anaemia and then mild anaemia. With HbSC individuals, those with moderate anaemia had the highest platelet count followed by mild anaemia and severe anaemia. Practically, more females had higher platelet count in relation to haemoglobin concentration than males, but the study did not look at the effect of gender on haemoglobin which can be said to be a limitation [42].

## 5. Conclusions

This study found that SCD patients have prolonged APTT, PT and an increased platelet count compared to non-SCD individuals, which might predispose them to bleeding and thrombocytosis. Significant difference was found between severe anaemia and the mean PT, mean APTT and mean platelet count, presupposing that a lower haemoglobin level might lead to an increase in platelet count and prolonged PT and APTT in SCD patients, particularly those with HbSS. It can therefore be concluded that individuals with SCD have prolonged PT, APTT and increased platelet count. This study has therefore thrown more light on the coagulation pattern seen in SCD patients by showing that a significant difference exists between PT, APTT and platelets in HbSS patients compared to HbSC patients and HbAA controls. Elongated PT, APTT and increased platelets may subsequently play a role in SCD complications. The findings also suggest that HbSC is not merely a milder form of HbSS and as such both diseases should be seen as different entities with regards to approaches for treatment and management. 

## Figures and Tables

**Table 1 diseases-07-00059-t001:** Distribution of demographic and clinical variables of study participants.

	HbAA	HbSS	HbSC
**Demographics**	Frequency	Frequency	Frequency
**Age**	23.63 ± 2.92	25.93 ± 10.96	35.56 ± 15.72
**Gender** Male	53(22.8%)	34(14.7%)	25(10.8%)
Female	18(7.8%)	61(26.2%)	41(17.7%)
Total	71(30.6%)	95(40.9%)	66(28.5%)
**Clinical**			
Temperature (mean ± SD) (°C)	36.9 ± 0.5	37.9 ± 1.3	37.3 ± 1.0
Pallor (%)	1(0.4%)	35 (15.1%)	18 (7.8%)
Fatigue	1(0.4%)	41 (17.7%)	26 (11.2%)
Pain crisis	0	49 (21.1%)	10 (4.3%)
Dactylitis	0	15 (6.5%)	2 (0.8%)
Frequent infections	0	16 (6.9%)	8 (3.4%)

**Table 2 diseases-07-00059-t002:** Distributions of haemoglobin levels among the various genotypes.

	HbSS	HbSC	HbAA (Control)	Total
**Hb(g/dL)**	Frequency	Frequency	Frequency	Frequency
<8.0 (severe anaemia)	49(30.4%)	2(1.2%)		51(31.7%)
8.0–10.9 (moderate anaemia)	42(26.1%)	32(19.9%)		74 (45.9%)
11.0–11.9 (mild anaemia)	2(1.2%)	17(10.6%)	7 (9.9%)	26 (9.9%)
**Coagulation parameters**	**Mean ± SD**	**Mean ± SD**	**Mean ± SD**	
PT (sec)	16.12 ± 0.6	15.39 ± 2.9	12.98 ± 1.22	14.83 ± 2.1
APTT (sec)	42.81 ± 1.61	46.90 ± 3.4	28.83 ± 3.96	32.84 ± 2.9
Platelets (10^9^/L)	439.65 ± 183.13	325.43 ± 114.3	251.96 ± 18.39	339.01 ± 105.27

**Table 3 diseases-07-00059-t003:** Comparison of the mean PT, APTT and platelet count and anaemia categories of participants with HbSS and HbSC.

HbSS–Hb (g/dL)	PT(sec) Mean ± SD	p-Value	APTT(sec) Mean ± SD	p-Value	PLATELET (10^9^/l) Mean ± SD	p-Value
<8.0 (severe anaemia)	18.93 ± 1.62	0.039*	43.00 ± 6.68	0.041*	506.14 ± 171.14	0.010*
8.0–10.5 (moderate anaemia)	15.5 ± 1.62	0.645	32.90 ± 5.77	0.719	539.83 ± 219.72	0.025*
11.0–11.9 (mild anaemia)	13.95 ± 0.49	0.895	30.53 ± 0.75	0.815	273 ± 30.41	0.495
**HbSC–Hb (g/dL)**						
<8.0 (severe anaemia)	16.35 ± 2.33	0.580	41.65 ± 8.98	0.044*	275.50 ± 102.53	0.502
8.0–10.5 (moderate anaemia)	14.64 ± 1.35	0.596	33.88 ± 5.95	0.746	381.56 ± 155.40	0.635
11.0–11.9 (mild anaemia)	15.18 ± 1.75	0.614	33.19 ± 6.79	0.690	319.24 ± 91.99	0.521

*Significant at *p* < 0.05.

**Table 4 diseases-07-00059-t004:** Post-hoc analysis of the mean difference of the parameters between the genotypes of cases and controls.

Bonferroni			Mean Difference	
Dependent Variable	(I) Geno	(J) Geno	(I-J)	*P*-Value
**Hb(g/L)**	AA	SS	5.428(0.228)	0.000**
	SC	AA	–2.606(0.248)	0.000**
	SC	SS	2.822(0.233)	0.000**
**PT(sec)**	AA	SS	–2.713(0.234)	0.001**
	SC	AA	1.900(0.255)	0.000**
	SC	SS	–0.812(0.239)	0.002**
**APTT (sec)**	AA	SS	–3.476(0.904)	0.000**
	SC	AA	4.889(0.986)	0.000**
	SC	SS	1.413(0.924)	0.382
**Platelet (×10^9^/L)**	AA	SS	–262.832(23.804)	0.000**
	SC	AA	107.012(25.944)	0.000**
	SC	SS	–155.820(24.314)	0.001**

**Significance at *p* < 0.05.

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
