# Peer review of "Severity of Anaemia Has Corresponding Effects on Coagulation Parameters of Sickle Cell Disease Patients"

_diseases, 2019, doi:10.3390/diseases7040059_

Round 1
Reviewer 1 Report
Dear authors,
Thank you for submitting the manuscript to the journal.
After careful reading, the manuscript is generally interesting with special focus on anemia. However, I would strongly recommend authors to relook at Discussion and improve further. Comments are in the attached file for your consideration.
Sincerely,

Reviewer 2 Report
This is an interesting paper that examines the abnormal coagulation profile of sickle cell disease as a correlate of anemia. A strength is the sample size.
Please add more detail about how the people were recruited for the blood samples - baseline status? no anticoagulant medications? any recent blood transfusions? hydroxyurea that could lower the platelet count? Known liver failure that would affect production of clotting factors? known renal failure that would affect hematocrit?
Table 1 shows an imbalance of gender in the different groups: more males in the HbAA group and more females in the HbSS and HbSC groups. This gender difference is a potential confounder for the differences in lab values. Males have higher hemoglobin levels. Pre-menopausal females might have less procoagulant tendency than males or post-menopausal females. The authors acknowledge that some data they publish did not have adkolewklfef
A previous paper examined the coagulation profile in sicklecell disease and should be cited for a complete view of the literature context.
1: Roeloffzen WW, Kluin-Nelemans HC, Mulder AB, Veeger NJ, Bosman L, de Wolf JT.
In normal controls, both age and gender affect coagulability as measured by
thrombelastography. Anesth Analg. 2010 Apr 1;110(4):987-94. doi:
10.1213/ANE.0b013e3181d31e91. PubMed PMID: 20357143.
2: Fourel V, Gabastou JM, Desroys du Roure F, Ehrhardt N, Robert A. Influence of age, sex and ABO blood group on activated partial thromboplastin time.
Haemostasis. 1993 Nov-Dec;23(6):321-6. PubMed PMID: 8034238.
3. Ajuwon MD1, Olayemi E2, Benneh AA3.Plasma levels of some coagulation parameters in steady state HBSC disease patients. Pan Afr Med J. 2014 Nov 17;19:289. doi: 10.11604/pamj.2014.19.289.4451. eCollection 2014.
Round 2
Reviewer 1 Report
Dear authors,
With this revised manuscript, it is suitable for publication with minor revision as shown in the text on page 7, lines 210-211 and 226-227.
Sincerely,

Author Response
The following changes have been made:
l has been changed to L in table 2. References 30, 31and 32 have been reset on page 7 line 210 - 211. References 35 and 36 have been reset on page line 217 and comment added. On page 7, line 226, the references 19 and 37 refers to the two works being referred to (Chinawa et al., (2013) and Nilesh et al., (2014) respectively).Thank you.